# Preparation and Performance of Micro-Arc Oxidation Coatings for Corrosion Protection of LaFe_11.6_Si_1.4_ Alloy

**DOI:** 10.3390/ma17061316

**Published:** 2024-03-13

**Authors:** Ruzhao Chen, Bin Fu, Jie Han, Hu Zhang, Ping Wang, Hongxia Yin

**Affiliations:** 1School of Material Science and Engineering, Tianjin University of Technology, Tianjin 300384, China; chenruzhao2020@126.com (R.C.); fubin@tjut.edu.cn (B.F.); 2Key Laboratory of Display Materials and Photoelectric Devices, Ministry of Education, Tianjin University of Technology, Tianjin 300384, China; 3College of Science, Tianjin University of Technology, Tianjin 300384, China; 4School of Materials Science and Engineering, University of Science and Technology Beijing, Beijing 100083, China; 5School of Material Science and Engineering, Yantai Nanshan University, Yantai 265713, China; wangping87881904@163.com (P.W.); yinhongxia_816@163.com (H.Y.)

**Keywords:** La-Fe-Si alloys, magnetic refrigeration materials, micro-arc oxidation, microstructure, corrosion resistance

## Abstract

The microstructure, corrosion resistance, and phase-transition process of micro-arc oxidation (MAO) coatings prepared on LaFe_11.6_Si_1.4_ alloy surfaces in different electrolyte systems were systematically investigated. Research has demonstrated that various electrolyte systems do not alter the main components of the coatings. However, the synergistic action of Na_2_CO_3_ and Na_2_B_4_O_7_ more effectively modulated the ionization and chemical reactions of the MAO process and accelerated the formation of *α*-Al_2_O_3_. Moreover, the addition of Na_2_CO_3_ and Na_2_B_4_O_7_ improved the micromorphology of the coating, resulting in a uniform coating thickness and good bonding with the LaFe_11.6_Si_1.4_ substrate. The dynamic potential polarization analysis was performed in a three-electrode system consisting of a LaFe_11.6_Si_1.4_ working electrode, a saturated calomel reference electrode, and a platinum auxiliary electrode. The results showed that the self-corrosion potential of the LaFe_11.6_Si_1.4_ alloy without surface treatment was −0.68 V, with a current density of 8.96 × 10^−6^ A/cm^2^. In contrast, the presence of a micro-arc electrolytic oxidation coating significantly improved the corrosion resistance of the LaFe_11.6_Si_1.4_ substrate, where the minimum corrosion current density was 1.32 × 10^−7^ A/cm^2^ and the corrosion potential was −0.50 V. Similarly, after optimizing the MAO electrolyte with Na_2_CO_3_ and Na_2_B_4_O_7_, the corrosion resistance of the material further improved. Simultaneously, the effect of the coatings on the order of the phase transition, latent heat, and temperature is negligible. Therefore, micro-arc oxidation technology based on the in situ growth coating of the material surface effectively improves the working life and stability of La(Fe, Si)_13_ materials in the refrigeration cycle, which is an excellent alternative as a protection technology to promote the practical process of magnetic refrigeration technology.

## 1. Introduction

Magnetic cooling technology is based on the magnetocaloric effect (MCE) [1,2], which refers to the magnetic entropy change (Δ*S*_M_) or temperature change (Δ*T*_ad_) produced by the application or removal of a magnetic field in adiabatic (or isothermal) conditions [3]. Compared to conventional cooling technology, it is highly efficient, environmentally friendly, stable, and reliable. Therefore, it is a novel and highly promising refrigeration technology. As the core of magnetic refrigeration technology, research regarding magnetic refrigeration working media is critical. Thus far, magnetocaloric materials with large MCEs near room temperature, such as Gd_5_Si_4−*x*_Ge*_x_* [4], La(Fe, Si)_13_ [5], and MnFeP*_x_*As_1-*x*_-based compounds [6], as well as certain rare earth-based oxides, alloys, and microwires [7], have been explored. Among these, La(Fe, Si)_13_-based materials with NaZn_13_ crystals are considered one of the most feasible alternative refrigerants for magnetic refrigeration owing to their significant advantages of having large MCEs, relatively high cooling power, and non-toxic constituent elements [8]. In addition, La(Fe, Si)_13_-based compounds have adjustable Curie temperatures (*T*_C_). However, as a typical first-order phase-transition MCE material, the order of the magnetic phase transition of La(Fe, Si)_13_ has a strong relationship with the Si content [9]. When 1.2 ≤ *x* ≤ 1.6, LaFe_13-*x*_Si*_x_* compounds demonstrate intense magnetically elastic coupling characterized by a lager negative expansion of the lattice at the Curie temperature and a magnetic-field-induced itinerant-electron metamagnetic transition (IEMT) [10]. Concurrently, the large changes in the magnetization and lattice parameters near the Curie temperature cause the La(Fe, Si)_13_ series of compounds to present significant magnetocaloric properties, and a small change in the magnetic entropy versus temperature curve [11].

Heat exchange is an important component in the working process of a magnetic refrigerator. Owing to their large heat capacity and high thermal conductivity, aqueous solutions or water-based fluids are utilized as highly efficient heat-exchange media between the magnetic refrigerants and loads for magnetic refrigerators operating near room temperature [12]. Notedly, LaFe_13−*x*_Si*_x_* compounds severely corrode in water without protection, which not only diminishes the working efficiency of the refrigerator but also damages its stability and working life [13]. In many cases, corrosion degradation deteriorates the magnetocaloric effect of the material, and the corrosion products affect the fluidity and heat conduction of the heat-transfer medium. Therefore, corrosion protection must be considered in practical processes. The causes of La(Fe, Si)_13_ material corrosion were investigated considering two perspectives. On the one hand, they contain the highly chemically active rare-earth element La, and on the other hand, the multi-phase structure and high potential difference between the phases caused by the non-equilibrium solidification of the La(Fe, Si)_13_ material weakens the electrochemical corrosion of the magnet when contacting the heat-exchange fluid [14]. The corrosion mechanism of the La(Fe, Si)_13_-based alloy principally depends on its multi-phase structure, and a micro galvanic couple is formed between two adjacent phases to accelerate the corrosion rate [15].

Thus far, several theoretical studies regarding corrosion behavior have been conducted considering the constituent elements, chemical composition, and original microstructure. By studying the corrosion resistance of annealed LaFe_11.6_Si_1.4_B*_y_* series alloys, Fe_2_B was found to replace *α*-Fe in the second phase after B doping, the difference in the micro galvanic corrosion potential between the second and matrix phases decreased, as well as the mixed current density, which inhibited the corrosion of the main phase [16]. Similarly, research regarding the latent heat behavior of La(Fe, Mn, Si)_13_ in the thermal cycle has demonstrated that Mn-doping can reduce the corrosion potential difference and strengthen the corrosion resistance of the alloy [17]. Additionally, the effect of adding Co and C was investigated, which concluded that the addition of Co and C can significantly enhance the membrane impedance and promote the formation of a protective corrosion membrane layer [14]. In addition, the non-stoichiometric La(Fe, Si)_13_ base alloy can also improve the corrosion resistance [18] by increasing the content of the specific impurity phases to amplify the magnetic and corrosion properties of the alloys.

In addition to the composition and inherent structural improvements, a utilization strategy for magnetocaloric materials coated with corrosion-resistant materials in refrigeration units was proposed, which inspired another study that revealed the corrosion behavior of LaFe_11.5_Si_1.5_/Cu compounds and found that Cu cladding separates the magnetic refrigeration working medium and corrosion liquid, effectively heightening the corrosion potential, reducing the corrosion current density, and improving the corrosion resistance of the alloy [19]. However, to ensure an efficient heat exchange with the heat-transfer liquid in AMR cycles, the materials must be processed into large surface areas. The application of cladding materials remains difficult, owing to several issues such as being limited to simple molded parts, poor cladding tightness, and the plating-liquid contamination of the environment [20]. The ion-implantation technology is not dependent on the silhouette. By using Cu ion implantation in the LaFe_11.6_Si_1.4_ alloy, the results indicated that the corrosion potential increased and the corrosion current decreased with Cu-ion implantation in the LaFe_11.6_Si_1.4_ alloy [21]. Nevertheless, the copper-formed surface corrosion layer is not as dense as those of other metal oxides, resulting in uneven and loose phenomena that accelerate corrosion. Recently, micro-arc oxidation (MAO) technology has been widely used in the anti-corrosion process of medical magnesium–zinc alloys owing to their dense and uniform coating, which has a good anti-corrosion performance, simple operation, no pollution, and no strict requirements for the appearance of the matrix materials [22]. At present, micro-arc oxidation technology is mainly applied to metals such as Mg, Al, and Ti, rather than Fe-based alloys. The main reason is that the MAO coating directly formed on the Fe substrate is not stable enough to achieve good corrosion resistance. In the trial experiment of the micro-arc oxidation of an iron-based rare-earth La(Fe, Si)_13_ alloy, it was found that the addition of rare-earth elements significantly improved the performance of MAO coatings on this series of alloys. If suitable electrolytes can be used, it is expected to obtain MAO coatings with good corrosion resistance. Therefore, the study of La(Fe, Si)_13_ series alloy MAO coatings is not only beneficial for the practical application of magnetic refrigeration technology, but also can promote the application of this coating on Fe-based alloys. However, there are no current studies regarding them.

In this study, a LaFe_11.6_Si_1.4_ alloy was used as the substrate of micro-arc oxidation. According to previous studies, the main component of the aluminate electrolyte system is Al_2_O_3_, which has a high strength, hardness, and abrasiveness, whereas the coating formed by it is compact and uniform, has a good thermal conductivity, and high corrosion resistance. Therefore, three electrolyte solutions based on the aluminate electrolyte system were designed using NaAlO_2_, Na_2_CO_3_, NaH_2_PO_4_, and Na_2_B_4_O_7_. The micro morphology of the MAO coating and its influence on the corrosion resistance and phase-transition process of LaFe_11.6_Si_1.4_ materials were systematically studied. This study introduces the foundation for preparing in situ MAO metal coatings on the surfaces of La(Fe, Si)_13_ magnetothermal materials.

## 2. Experimental

LaFe_11.6_Si_1.4_ alloys were prepared by arc-melting pure La (99.9%), Fe (99.9%), and Si (99.9%) in a high-purity argon atmosphere. Cylinder samples of Φ8 × 3 (±0.1) mm in size were cut out of the ingot. The samples were annealed at 1323 K for 12 days in a quartz tube filled with an argon atmosphere, followed by ice-water quenching. The oxides on the surface of the sample were removed using a 1000-mesh silicon carbide sandpaper. Surface grease was then removed by ultrasonic cleaning. In the MAO process, the LaFe_11.6_Si_1.4_ sample was the working anode and the stainless steel was the cathode. The voltage and pulse frequency were set at 500 V and 1200 Hz, respectively, and the treatment duration was 20 min. The electrolyte temperature was maintained at 20 °C by circulating the water-cooling bath. The main electrolytes used for MAO were NaAlO_2_, Na_2_CO_3_, NaH_2_PO_4_, and Na_2_B_4_O_7_. Table 1 and Figure 1 list the combinations of electrolytes and the schematic diagram of the MAO device used in this experiment; the coatings prepared are represented by M1, M2, and M3, respectively.

The microstructures and elemental compositions of the coatings were analyzed using scanning electron microscopy (SEM, JSM-6360LV, JEOL, Tokyo, Japan) and energy-dispersive spectrometry (EDS, Oxford Atec X-max 50, Oxford Company, Oxford, UK). X-ray diffraction (XRD, Ultima-IV, Rigaku, Tokyo, Japan) was used to study the crystal structure of the samples. The porosity of the coating surface was quantified using ImageJ software (ImageJ 1.8.0, National Institutes of Health, Bethesda, MD, USA). The coating-binding force was tested using a WS-2005 coating adhesion scratch instrument (Shanghai Shenrui Instrument Co.,Ltd., Shanghai, China). 

Dynamic polarization (IE) testing of the coatings was performed using a standard three-electrode electrochemical analyzer/workstation (VersaSTAT MC, Ametek Company, Berwyn, PA, USA). For the electrochemical study, each measurement was performed in a standard three-electrode cell consisting of a LaFe_11.6_Si_1.4_ working electrode, saturated calomel reference electrode (SCE), and platinum counter electrode. Distilled water was used as the test solution in the 0.2 cm^2^ test area. The polarization curves of the samples in the test solution were recorded at a scanning speed of 1 × 10^−3^ V/s. The thermal cycle was conducted near *T*_C_ using differential scanning calorimetry (DSC, DSC214, NETZSCH-Gerätebau GmbH, Selb, Germany) at a scanning rate of 3 K/min.

## 3. Results and Discussion

Figure 2 presents the XRD patterns of the MAO-coated samples with different electrolyte compositions, which demonstrates that all the MAO coatings present characteristic peaks of *α*-Al_2_O_3_, *γ*-Al_2_O_3_, Fe_2_O_3_, FeAl_2_O_4_, and FePO_4_, indicating that different electrolyte compositions did not change the main components of the MAO coatings. Specifically, the diffraction peaks of Fe_2_O_3_ and FePO_4_ were observed at 45.5° and 65.0° [23], respectively, for all the samples. Notably, owing to the layered porous structure of the coating, X-rays can penetrate the coating onto the Fe_2_O_3_-dense oxide layer, resulting in the appearance of Fe_2_O_3_ diffraction peaks. This also indicates that the iron oxide on the substrate participates in the MAO reaction [23,24]. In addition, the FeAl_2_O_4_ diffraction peak was located at 45.5°, which overlaps with those of the first two Fe-containing compounds [24]. The diffraction peaks of *α*-Al_2_O_3_ were located at 31.7°, 37.2°, 44.6°, 60.3°, and 66.0°, and 43.3°, 47.0°, 56.3°, and 82.5° corresponded to *γ*-Al_2_O_3_. Compared with M1, the diffraction-peak intensity of Fe_2_O_3_ remarkably decreased in the M2 spectrum, whereas those of *α*-Al_2_O_3_ and *γ*-Al_2_O_3_ increased, indicating that the addition of Na_2_CO_3_ promotes the growth of the *α*-Al_2_O_3_ film layer and affects the reaction of MAO [25,26]. The diffraction peaks of Fe_2_O_3_ and FePO_4_ in M3 nearly disappeared, and the intensity of the *γ*-Al_2_O_3_ diffraction peaks decreased, indicating that the addition of the Na_2_B_4_O_7_ resolved the iron-oxide formation and significantly inhibited FePO_4_ formation in the envelope [27]. Generally, the type of Al_2_O_3_ crystal in the coating directly affects the corrosion resistance, and a higher content of *α*-Al_2_O_3_ improves the corrosion resistance.

Figure 3a–d, as well as Figure 3e,f, demonstrate the surface morphology of M1, M2, and M3 MAO coatings prepared under different electrolyte compositions, respectively. The content of each element obtained from the EDS analysis is listed in Table 2. Because it was performed in a constant pressure mode, the sample surfaces under different electrolytes exhibited “volcanic characteristics” of the MAO coating [28,29]. The mastoid morphology is the main surface feature that can be observed in the high magnification images, as shown in point A in Figure 3a, where the upper-right illustration presents an enlarged view of point A. All the coatings have black circular pores distributed in the molten region on the surface. These circular micropores are volcanic vents, which are residual channels of the discharge reaction, and are caused by the outflow of molten oxides from the discharge channel and are cooled by relatively cold electrolytes [30]. The microcracks in the coating are caused by the thermal stress generated by the rapid solidification of molten oxide [31]. The coating of numerous microcracks and microporous structures was clearly observed in M1, resulting in an uneven surface and higher roughness (Figure 3b). However, the appearance of cracks indicates that the formation process of the MAO coating is unstable during arc interruption, and it also suggests the presence of loose Fe_2_O_3_ and FePO_4_. Similarly, the pores and cracks as defects will be detrimental to the anti-corrosion performance of the material. M1 is mainly composed of Al, O, Fe, and P elements, with Al and Fe atoms accounting for 29.7% and 6.6%, respectively, as shown in Table 2. When Na_2_CO_3_ was added to M2, the small hole at the center of the circular surface protrusion was a typical charge perforation in the MAO process (Figure 3d). Na_2_CO_3_ has a positive impact on the formation of coatings, which is manifested by a significant reduction in the surface cracks, a lower roughness, and a denser coating. This phenomenon is due to the addition of Na_2_CO_3_, changing the solution environment and thus improving the discharge mechanism [25,26], making it more conducive to the production of Al_2_O_3_, resulting in a higher content of Al_2_O_3_ in the coating. The EDS analysis confirms that the content of Al and O slightly increased in M2, whereas the content of Fe and P decreased, indicating that the growth of the Al_2_O_3_ crystals effectively improved during the MAO process [32]. The surface morphology of M3 is generally consistent with that of M2, but the large pores are significantly reduced, shrinking to half. The coating surface is uniform and dense (Figure 3e). During the formation process of M3 coating, the addition of Na_2_CO_3_ and Na_2_B_4_O_7_ further improved the discharge mechanism of micro-arc oxidation [28], thereby changing the formation mechanism of the surface layer and leading to the disappearance of cracks and the sealing of most discharge holes. On the other hand, the presence of Na_2_B_4_O_7_ promoted the decomposition of iron oxide, making it more conducive to the generation of Al_2_O_3_. Therefore, the combined effect resulted in a smoother and more uniform surface layer of the M3 coating. According to the EDS results, the atomic percentages of Al and O are 36.9% and 58.2%, respectively, with a ratio of 0.634, which is notably close to the Al/O ratio of the Al_2_O_3_ compound (0.667). Overall, the addition of both Na_2_B_4_O_7_ and Na_2_CO_3_ regulates the reaction process, significantly inhibiting the oxidation reaction of Fe in the substrate and catalyzing the generation of Al_2_O_3_. Simultaneously, the appropriate amount of Na_2_CO_3_ or Na_2_B_4_O_7_ makes the coating surface uniform and delicate.

The cross-sectional SEM morphologies and EDS elemental line-scanning distribution curves of the MAO coatings in different electrolytes are shown in Figure 4. Figure 4a,c,e correspond to M1, M2, and M3, respectively. Specifically, Figure 4b,d,f correspond to the enlarged views of the region in the circle in Figure 4a,c,e, respectively, whereas a1, b2, and c3 correspond to the line scanning at the transverse line, respectively. Figure 4a demonstrates that the coating thickness is uniform, whereas Figure 4b indicates that the substrate material is closely connected to the coating; however, the presence of deeper cracks causes the coating surface to become dense and thus fall off. In addition, the line-scan pattern demonstrates that the diffraction peaks of Al and O are alternately serrated, whereas the diffraction peaks of Fe and P appear simultaneously. This indicates that the compositional distribution of the coating was uneven and that there were other oxides that significantly reduced the bonding strength. Furthermore, the average coating thickness of M1 was 40 ± 2 μm. The overall condition of M2 was good, but according to the energy spectrum, impurities remained and the coating thickness (33 ± 2 μm) was smaller than the other two coatings. The coating thickness of M3 reached 65 ± 2 μm, there was no gap with the substrate, and its cross-sectional state was sufficiently stable, dense, and smooth. A comparison of the thicknesses of the coatings is shown in the SEM images. Changes in the electrolyte composition can apparently result in significant differences in the cross-sectional morphologies of the coatings. Notedly, there was a stable Fe-A1 co-position transition zone between the coating and substrate with a thickness of 3 ± 0.5 μm, indicating that substrate oxidation occurs during the initial stage of the coating formation.

The surface pore distribution of the MAO coatings with different electrolytes is shown in Figure 5a–c, where a–c corresponds to M1–M3, respectively, and the porosity calculated using the ImageJ software is shown in Figure 5d. Overall, owing to the constant pressure mode, the dominant pore structure was the discharge channel [33]. M1 was mainly composed of micropores and cracks with a porosity of 15.28%. The addition of Na_2_CO_3_ to M2 significantly regulated the MAO-discharge process and improved the density of the membrane layer and growth of the outer layer, resulting in the formation of a porous morphology; its porosity decreased to 10.78%. When Na_2_B_4_O_7_ was added, the porosity did not change; however, the hole diameter significantly decreased. This is because the discharge process significantly improved, the conductivity was enhanced, and the solidified oxide melted and decomposed again, causing it to slightly solidify or become submerged when encountering colder electrolytes. In conclusion, an appropriate electrolyte composition can improve the pore morphology of the coating surface and make it smooth, compact, and uniform.

The bonding strength between the coatings and the La-Fe-Si substrate is critical for practical use. In this regard, Figure 6 presents the bonding-force analysis of the coatings prepared with different electrolyte components. According to this definition [34], the peak position of the first acoustic signal corresponds to the load capacity, that is, the bonding force. The ratio of the first load to the surface area of the sample is the bonding strength. Figure 6 demonstrates that M1 had the lowest adhesion among the three coatings, for which the bonding force was 10.83 ± 0.15 N and bonding strength was 86.25 ± 1.18 MPa. This is owing to the crack defects, an unstable Fe-Al transition layer, and a complex coating composition, resulting in a low bonding force. The bonding force of M2 was 16.65 ± 0.37 N, and the bonding strength was 132.56 ± 2.29 MPa. Compared with M1, the bonding capacity of M2 increased, which further verifies that the addition of Na_2_CO_3_ changes the discharge reaction and improves the distribution of the coating structure and diffusion process of the components. However, the bonding force and bonding strength of the M3 coating slightly decreased, which were 16.05 ± 0.44 N and 127.80 ± 3.54 MPa, respectively. This is owing to the influence of the discharge reaction on the M3 layer; furthermore, the increase in pores in the outermost membrane layer leads to a decrease in the density, and the stability of the Fe-A1 transition layer remains unchanged.

Figure 7 presents the polarization curves (PDP) obtained from the kinetic potential polarization tests of the samples prepared in different electrolytes in distilled water. The dynamic potential polarization analysis was performed in a three-electrode system consisting of a LaFe_11.6_Si_1.4_ working electrode, a saturated calomel reference electrode and a platinum auxiliary electrode. The electrochemical data obtained after Tafel fitting are shown in Table 3. The samples without coatings demonstrated poor corrosion resistance, whereas those with coatings had significantly improved corrosion resistance. Among these, the M3 coating exhibited the best protective performance, with a polarization potential and corrosion current density of −0.50 V and 1.32 × 10^−7^ A/cm^2^, respectively, which was owing to the large coating thickness and small pore size, inhibiting the corrosion of the sample. The corrosion potential apparently increased from −0.68 V of LaFe_11.6_Si_1.4_ to −0.50 V of M3, and the corrosion current density decreased from 8.96 × 10^−6^ A/cm^2^ to 1.32 × 10^−7^ A/cm^2^, indicating an increase in the corrosion potential and decrease in the corrosion current density of the sample. This also confirms that the addition of Na_2_CO_3_ and Na_2_B_4_O_7_ improved the microstructure of the coating and enhanced the corrosion resistance of the substrate material.

The aforementioned results were obtained owing to the following: ① The coating has a higher *α*-Al_2_O_3_ content, and its good coating performance separates the substrate from the solution, inhibits the corrosion of the substrate, and decelerates the electrochemical corrosion. ② Owing to the unique, uniform, and dense micromorphology of the constant-voltage discharge mode, contact with the heat-exchange medium is effectively avoided. ③ Owing to the high bonding strength of the coating, the dense layer and substrate were directly connected.

As indicated above, the coating was successfully prepared on the surface of the LaFe_11.6_Si_1.4_ alloy, and the corrosion resistance of the substrate was effectively improved. Owing to the need for heat exchange between the magnetic refrigeration materials and the medium during the operation, it is necessary to consider whether the coating affects the heat absorption and release during the phase-transition process. The M3 coating demonstrated an excellent corrosion resistance, as shown in Figure 8, which presents the DSC curves of the samples without the surface treatment and M3-coated samples; here, the peak represents the heat absorbed by the material during the phase-transition process. The phase-transition temperatures, determined by differentiating the DSC (DDSC) curves [35], were 190 K and 187 K, respectively. Endothermic peaks occurred at the same temperature during the thermal cycling process, indicating that the phase transition temperature was nearly consistent during the cycling process [36,37]. The phase-transition latent heat indicates the integral of the DSC curve peak over time [38]:(1)$$\Delta H=\int \frac{dQ}{dt}\cdot \;dt$$

In the formula, $\frac{dQ}{dt}$ is the heat-flux rate, expressed as the difference in power delivered to the sample and the reference while keeping their temperatures equal (where *Q* and *t* are heat and time, respectively). The $\int \frac{dQ}{dt}\cdot \;dt$ is the latent heat, which is directly calculated by the built-in software of the DSC equipment (NETZSCH Proteus 5.0, NETZSCH-Gerätebau GmbH, Selb, Germany). The calculations determined that the values without and with the coating were 4.43 J/g and 4.26 J/g, respectively. The phase-change latent heats of the uncoated and coated samples were nearly identical. In addition, the sharp latent thermal peak indicates that the coated LaFe_11.6_Si_1.4_ remains a first-order magnetic-phase transition material. Therefore, the micro-arc oxidation technology not only improves the corrosion resistance of the material, but also does not alter the other properties, which is critical for protecting the substrate from damage.

## 4. Conclusions

In this study, the micromorphology, coating performance, corrosion resistance, and magnetocaloric effect of MAO coatings prepared on LaFe_11.6_Si_1.4_ alloy surfaces using different electrolyte systems were systematically compared and analyzed. X-ray diffraction mapping revealed that the different electrolyte systems had little effect on the main components of the coating; however, there were apparent differences in the Al_2_O_3_ content. The addition of Na_2_CO_3_ and Na_2_B_4_O_7_ significantly improved the MAO-discharge process, making the coating structure and morphology on the LaFe_11.6_Si_1.4_ alloy surface more compact and smoother, with a more uniform thickness and improved performance. Furthermore, the addition of Na_2_CO_3_ and Na_2_B_4_O_7_ reduced the surface porosity and pore diameter of the LaFe_11.6_Si_1.4_ alloy while improving the binding force and binding strength between the coating and substrate, which were 16.65 ± 0.37 N and 132.56 ± 2.29 MPa, respectively. Compared with the alloy without the surface treatment, the corrosion resistance of the LaFe_11.6_Si_1.4_ alloy with the MAO coating was significantly improved, and the self-corrosion current density of the coating was reduced to 1.32 × 10^−7^ A/cm^2^. In addition, the MAO coating enhanced the corrosion resistance without affecting the magnetothermal properties of the LaFe_11.6_Si_1.4_. Therefore, micro-arc oxidation, a technology that decelerates the corrosion of the substrate, effectively provides corrosion protection for La(Fe, Si)_13_ in refrigeration cycles and is expected to promote the practicality of magnetic refrigeration technology.

## Figures and Tables

**Figure 1 materials-17-01316-f001:**
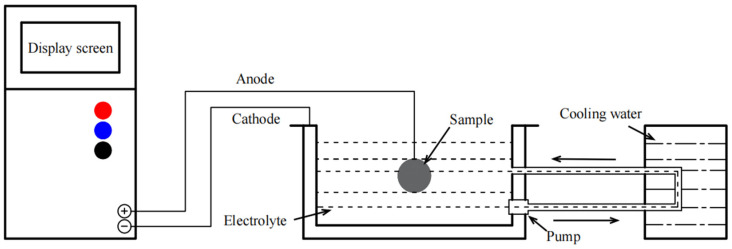
Schematic diagram of the micro-arc oxidation device.

**Figure 2 materials-17-01316-f002:**
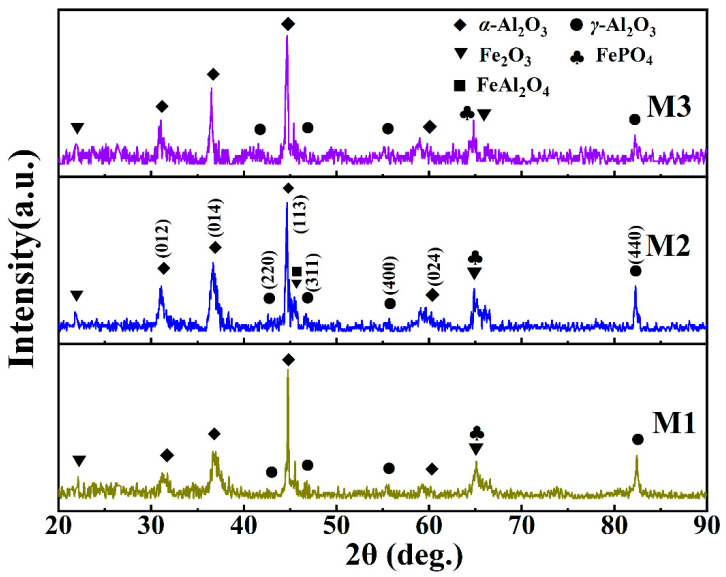
XRD pattern of the MAO coating under different electrolyte components.

**Figure 3 materials-17-01316-f003:**
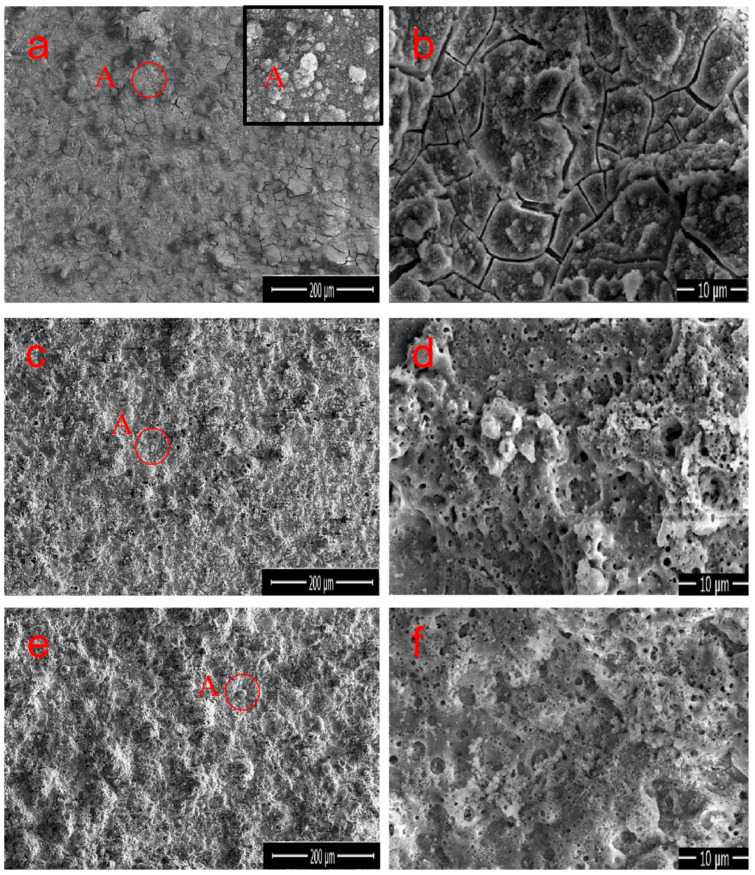
SEM images of the (**a**,**b**) M1, (**c**,**d**) M2, and (**e**,**f**) M3 MAO coatings under different electrolyte compositions. The right images correspond to enlarged views of area A in the left images.

**Figure 4 materials-17-01316-f004:**
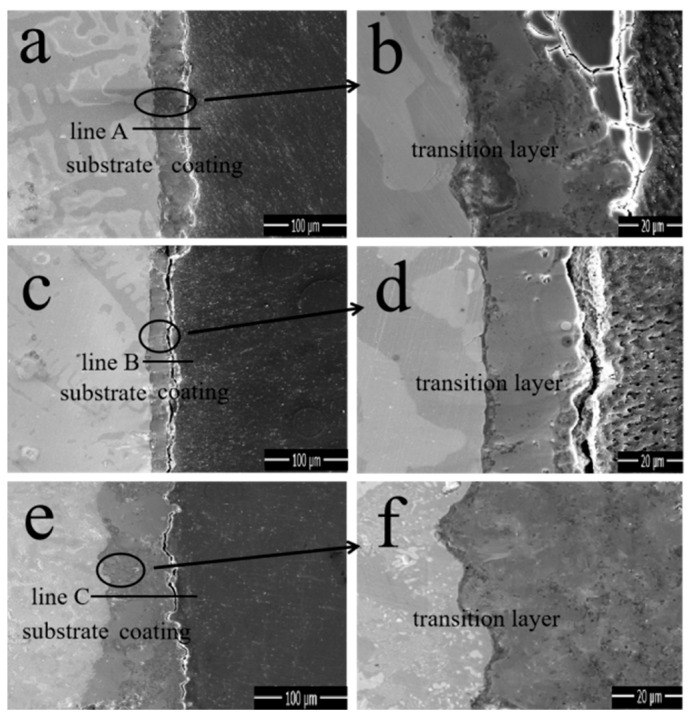
SEM image and EDS line-scan distribution curve of the MAO coating cross-section: (**a**,**b**,**a1**) M1, (**c**,**d**,**b2**) M2, (**e**,**f**,**c3**) M3.

**Figure 5 materials-17-01316-f005:**
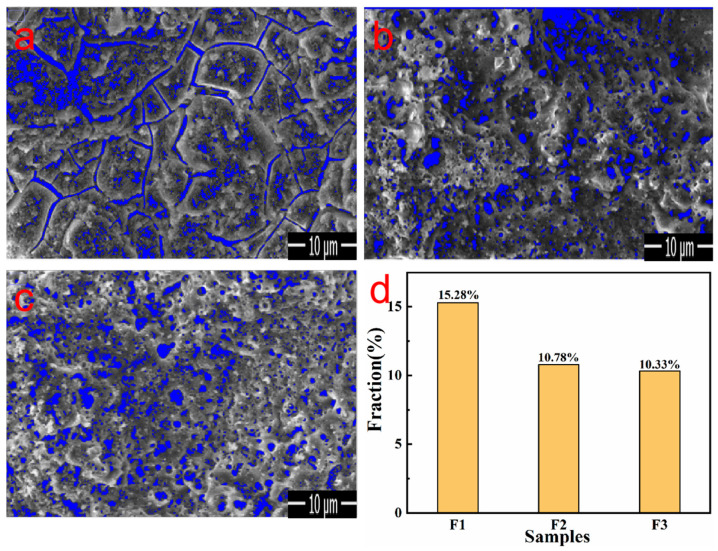
Pore images of the MAO coatings under different electrolyte components (**a**–**c**); porosity bar chart (**d**).

**Figure 6 materials-17-01316-f006:**
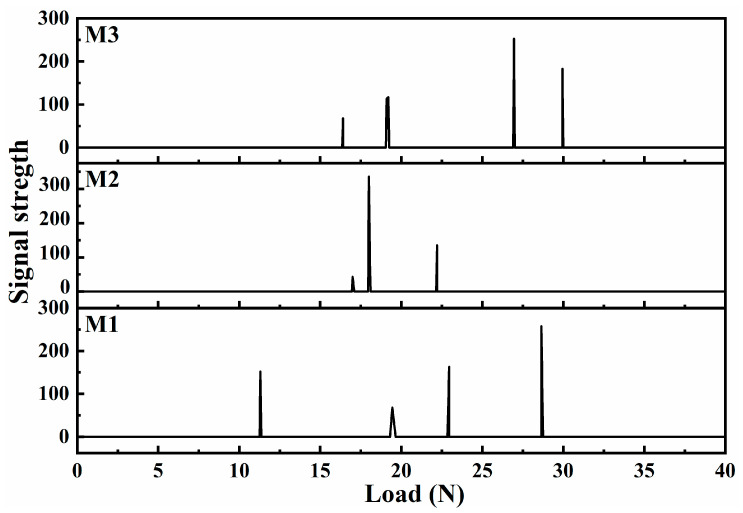
Adhesion of the MAO coatings under different electrolyte compositions.

**Figure 7 materials-17-01316-f007:**
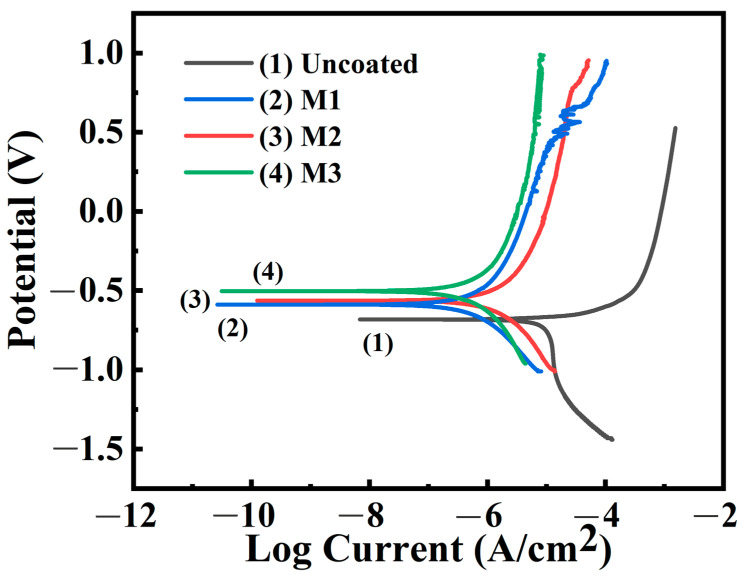
Polarization curves of the MAO coatings under different electrolyte compositions.

**Figure 8 materials-17-01316-f008:**
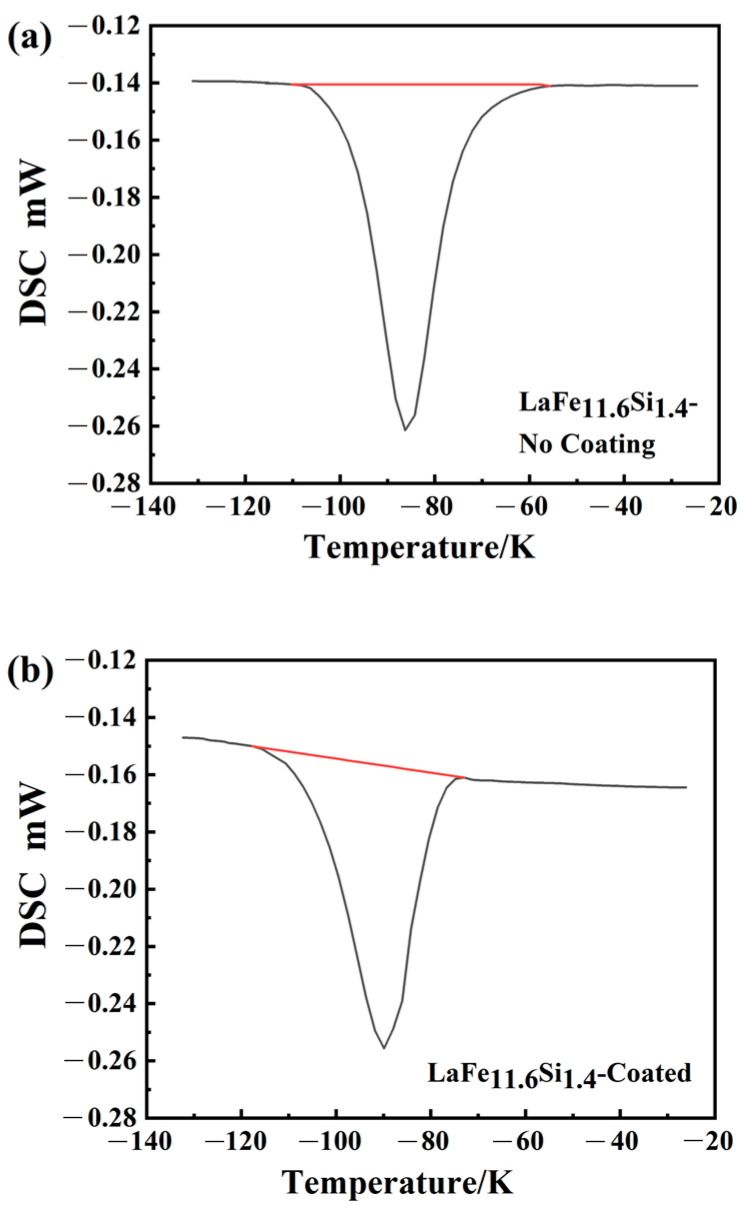
DSC curves of the LaFe_11.6_Si_1.4_ compound (**a**) without coating and (**b**) with coating.

**Table 1 materials-17-01316-t001:** Composition and concentration of the electrolytes used in different MAO coatings.

Coatings	NaAlO_2_ (g/L)	NaH_2_PO_4_ (g/L)	Na_2_CO_3_ (g/L)	Na_2_B_4_O_7_ (g/L)
M1	15	3	/	/
M2	15	3	3	/
M3	15	3	3	3

**Table 2 materials-17-01316-t002:** The elemental content (atomic percentage) of the MAO coatings under different electrolyte components.

Sample	Al (%)	O (%)	P (%)	Fe (%)
M1	29.7	59.9	3.9	6.6
M2	33	60.4	3.1	3.6
M3	36.9	58.2	2.3	2.7

**Table 3 materials-17-01316-t003:** MAO coating data fitted by the Tafel polarization curve under different electrolyte compositions.

Sample	Corrosion Current Density i_corr_ (A/cm^2^)	Corrosion Potential E_corr_ (V)	Corrosion Rate CR (mm/y)
LaFe_11.6_Si_1.4_	8.96 × 10^−6^	−0.68	0.0690
M1	3.37 × 10^−7^	−0.56	0.0026
M2	1.51 × 10^−7^	−0.53	0.0011
M3	1.32 × 10^−7^	−0.50	0.0010

## Data Availability

Data are contained within the article.

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
