# Peer review of "Preparation and Performance of Micro-Arc Oxidation Coatings for Corrosion Protection of LaFe11.6Si1.4 Alloy"

_materials, 2024, doi:10.3390/ma17061316_

Round 1
Reviewer 1 Report
Comments and Suggestions for Authors
The work is devoted to the micro-arc oxidation coatings for corrosion protection of the LaFe11.6Si1.4 alloy.
The topic may be of interest to scientific community, so in my opinion the work is worth of publication.
The manuscript is well organized, well written and readable. The work is sufficiently detailed and conclusive.
The article needs minor revision:
1) Abstract:
When providing the potential values, you must also provide the reference electrode.
2) Figure 3 and Figure 4:
The title of Fig.3 should include information about what each photo a-f demonstrates.
The same note applies to Fig.4.
3) Table 2:
- For corrosion rate, the unit description mm/y is commonly used, not mm/a.
- There is no description of what I0 and E0 are.
- I suggest using the designations Icorr and Ecorr instead of I0 and E0.
4) DSC curve peak:
What do the individual symbols in the given formula mean?
Author Response
Dear Editors and Reviewers:
Thank you for your letter and for the reviewers’ comments concerning our manuscript. All the comments are valuable and very helpful for revising and improving our paper, as well as the important guiding significance to our researches. We have addressed all the comments carefully and have made answers, supplements and corrections. In addition, some errors in the references have been corrected. All revisions are highlighted in red font in the manuscript. The responds to the reviewer’s comments are as following:
Comment 1:
Abstract:
When providing the potential values, you must also provide the reference electrode.
Response: Thank you very much for pointing out the problem. According to this comment, we have provided an explanation of the electrode system in the Abstract section.
Comment 2:
Figure 3 and Figure 4:
The title of Fig.3 should include information about what each photo a-f demonstrates.
The same note applies to Fig.4.
Response: Thank you very much for pointing out these problems! According to this opinion, we have made modifications to the titles of Fig.3 and Fig.4.
Comment 3:
Table 2:
- For corrosion rate, the unit description mm/y is commonly used, not mm/a.
- There is no description of what I0 and E0 are.
- I suggest using the designations Icorr and Ecorr instead of I0 and E0.
Response: When reviewing this comment, we found that we mistakenly labeled Table 3 as Table 2. We have corrected this error and sincerely apologize for making such a low-level mistake. In accordance with this comment, the unit and designations have been modified accordingly, and the descriptions of Icorr and Ecorr have been also marked in this table. Thank you for these professional suggestions!
Comment 4:
DSC curve peak:
What do the individual symbols in the given formula mean?
Response: Thank you for your reminder. We did overlook the annotation of the symbols. Meanwhile, due to this opinion, we feel that it is necessary to add explanations for the entire formula and methods of data acquisition, so that readers can have a clearer understanding. Therefore, we have added corresponding explanations in the text, including explanations of symbols in the formula.
Reviewer 2 Report
Comments and Suggestions for Authors
The manuscript titled "Preparation and performance of micro-arc oxidation coatings for corrosion protection of the LaFe11.6Si1.4 alloy" is a detailed study and can potentially be a good fit for publication in Materials but the authors would need to address to following concerns before publication:
-
1. How do MAO coatings compare with other surface treatment technologies applied to LaFe11.6Si1.4 or similar alloys in terms of corrosion resistance, microstructure stability, and impact on magnetocaloric properties? A comparative analysis could provide a clearer understanding of the advantages or unique benefits of MAO coatings over other coatings.
-
2. Can you elaborate on the mechanisms by which the addition of Na2CO3 and Na2B4O7 to the electrolyte improves the corrosion resistance of the MAO coatings? Understanding the chemical or physical interactions at play could provide deeper insights into the process optimization.
-
3. The manuscript mentions that the addition of Na2CO3 and Na2B4O7 results in a more uniform coating thickness. How does the thickness of the MAO coating influence the corrosion resistance, micromorphology, and magnetocaloric effect? Is there an optimal thickness range that balances these properties effectively?
-
4. Have long-term stability and durability tests been conducted on the MAO-coated LaFe11.6Si1.4 alloy? Information on the coating's performance over time and under various environmental conditions would be beneficial for assessing its practical application in magnetic refrigeration technology.
-
5. Could you discuss the economic viability and environmental impact of using MAO coatings, especially with the addition of Na2CO3 and Na2B4O7, in commercial magnetic refrigeration applications? An analysis of the cost-effectiveness and any potential environmental concerns associated with the electrolyte components would be valuable.
-
6. What challenges, if any, do you anticipate in scaling up the MAO coating process for larger-scale applications or industrial manufacturing? Addressing potential scalability issues or limitations could help in understanding the feasibility of applying this technology in the real world.
-
7. Based on your findings, what are the next steps in research that could further optimize the MAO coating process or explore new applications of the coated LaFe11.6Si1.4 alloy? Suggestions for future studies could help in mapping out the progression of this research field.
Minor editing of English language required
Author Response
Dear Editors and Reviewers:
Thank you very much for taking the time to review our manuscript. All the comments are valuable and very helpful for revising and improving our paper, as well as the important guiding significance to our researches. We have addressed all the comments carefully and have made answers, supplements and corrections. All revisions are highlighted in red font in the manuscript. The responds to the reviewer’s comments are as following:
Comment 1: How do MAO coatings compare with other surface treatment technologies applied to LaFe11.6Si1.4 or similar alloys in terms of corrosion resistance, microstructure stability, and impact on magnetocaloric properties? A comparative analysis could provide a clearer understanding of the advantages or unique benefits of MAO coatings over other coatings.
Response: Thank you for your question. In the preliminary research, we found that there are currently two main approaches to improving the corrosion resistance of La(Fe, Si)13 materials: Element doping and surface treatment. The former method has limited effectiveness, so in recent years researchers have proposed the method of cladding. But the application of cladding materials such as Cu remains difficult owing to several issues such as being limited to simple molded parts, poor cladding tightness, and plating liquid contamination of the environment. By this, we are trying to find better surface treatment solutions. The reason we attempt to prepare the MAO coatings on rare earth iron based alloys is the characteristics of this technology (good anti-corrosion performance, simple operation, no pollution, and no strict requirements for the appearance of the matrix materials) are highly suitable for the corrosion protection requirements of La(Fe, Si)13 magnetic refrigeration materials. For this reason, we conducted this exploratory experiment. Meanwhile, due to this review comment, we feel that the significance of this work was not clearly stated in manuscript. Therefore, in the introduction section of the revised manuscript, corresponding information have been added. Thank you for this important comment!
Comment 2: Can you elaborate on the mechanisms by which the addition of Na2CO3 and Na2B4O7 to the electrolyte improves the corrosion resistance of the MAO coatings? Understanding the chemical or physical interactions at play could provide deeper insights into the process optimization.
Response: Thank you for this comment. Based on experimental analysis, we believe that the better performance of the coating is due to two factors. On the one hand, the addition of Na2CO3 and Na2B4O7 improves the discharge mechanism of micro-arc oxidation. This factor changes the formation mechanism of the surface layer, leading to the disappearance of cracks, the sealing of most discharge holes, resulting in a smoother and more uniform surface layer. On the other hand, the addition of Na2CO3 changes the solution environment, and the addition of Na2B4O7 promoted the decomposition of iron oxide, making it more conducive to the generation of alumina, resulting in a higher alumina content in the coating. This factor greatly improves the corrosion resistance. The combined effect of the above two factors improves the performance of the coating. The logic was not expressed clearly in the original manuscript, so we made targeted modifications to the discussion of Fig. 3 and Table 2 in the revision.
Comment 3: The manuscript mentions that the addition of Na2CO3 and Na2B4O7 results in a more uniform coating thickness. How does the thickness of the MAO coating influence the corrosion resistance, micromorphology, and magnetocaloric effect? Is there an optimal thickness range that balances these properties effectively?
Response: Thank you for this comment. Due to the non-magnetic nature of the MAO coating, it will not have a significant impact on the magnetocaloric effect of the protected material. Uniform and dense with a certain thickness can ensure that the coating does not contain too many pores and cracks, thereby protecting the substrate material. Our idea is that, under the premise of providing corrosion protection, the coating should be as thin as possible, which can maximize the proportion of magnetic refrigeration materials per unit mass and preserve the magnetocaloric effect. From the comparison of the thickness, crack and through-hole distribution of the three MAO coatings, it can be seen that when the film thickness reaches about 60 μm, an ideal corrosion resistance effect can be achieved. But how to prepare a coating that is equally uniform, delicate, and thinner, further research may be needed on the optimization of the electrolyte solution.
Comment 4: Have long-term stability and durability tests been conducted on the MAO-coated LaFe11.6Si1.4 alloy? Information on the coating's performance over time and under various environmental conditions would be beneficial for assessing its practical application in magnetic refrigeration technology.
Response: Thank you for asking this question. We agree with this comment. Stability and durability are very important properties for all materials that need to be applied to magnetic refrigerators. Therefore, the same tests should also be conducted on the MAO coatings. But it must be admitted that we have not yet conducted such experiments. Because this is our first attempt to prepare MAO coatings on LaFe11.6Si1.4 alloy, we mainly focused on how to successfully prepare them, as well as some basic properties such as the microstructure and adhesion. But we are certain that stability and durability tests will be conducted in the following studies, such as multi cycle heating and cooling, magnetization and demagnetization coating stability testing, and long-term corrosion behavior testing. Thank you again for your guidance!
Comment 5: Could you discuss the economic viability and environmental impact of using MAO coatings, especially with the addition of Na2CO3 and Na2B4O7, in commercial magnetic refrigeration applications? An analysis of the cost-effectiveness and any potential environmental concerns associated with the electrolyte components would be valuable.
Response: Thank you very much for this comment. Actually, the micro-arc oxidation (MAO) technology is not a very new technology. It has been widely used in the anti-corrosion process of metals such as Mg, Al, Ti alloys. The large-scale practical application shows that the cost of this technology is controllable. The use of MAO in many medical alloys also proves that it is an environmentally friendly technology. An important reason why we attempted to prepare MAO coatings on LaFe11.6Si1.4 alloy is that compared to corrosion resistance schemes represented by Cu cladding, there is no problem of electroplating solution polluting the environment. Before the experiment, we also confirmed through preliminary investigation that Na2CO3 and Na2B4O7 are safe and inexpensive. Therefore, from an economic and environmental perspective, we are optimistic about the application of MAO technology in La(Fe, Si)13 series alloys.
Comment 6: What challenges, if any, do you anticipate in scaling up the MAO coating process for larger-scale applications or industrial manufacturing? Addressing potential scalability issues or limitations could help in understanding the feasibility of applying this technology in the real world.
Response: Thank you very much for this forward-thinking question. To be honest, we did not consider this important issue that is very closely related to practical applications. After careful consideration, we believe that there may be a problem that must be addressed in large-scale production: To meet the application needs of different temperature ranges and environments, the composition of La(Fe, Si)13 series material is very variable. Whether it is possible to use a limited number of universal MAO processes to meet the corrosion protection requirements of all La(Fe, Si)13 alloys with different compositions is a prerequisite that must be addressed in large-scale production. For example, if it has to frequently configure and replace the electrolyte for micro-arc oxidation, the production cost will be very high, leading to serious constraints on practicality. To solve such problems, further researches are necessary.
Comment 7: Based on your findings, what are the next steps in research that could further optimize the MAO coating process or explore new applications of the coated LaFe11.6Si1.4 alloy? Suggestions for future studies could help in mapping out the progression of this research field.
Response: Before reading your review comments, we plan to further study the following content:
- At present, MAO technology is mainly applied to metals such as Mg, Al, Ti, rather than Fe based alloys. The main reason is that the MAO coating directly formed on the Fe substrate is not stable enough to achieve good corrosion resistance. If it is necessary to prepare MAO thin films on Fe based alloys, a dense layer needs to be pre formed using hot dip aluminum method. This situation restricts the application of MAO in iron-based alloys. But we found that MAO coatings with good adhesion can be directly generated on LaFe11.6Si1.4 alloy. This is likely due to the presence of rare earth elements in La(Fe, Si)13 series alloys. So, the specific role and mechanism of rare earth elements in the entire process of micro-arc oxidation are worth further research.
- Thermal conductivity is an important property of magnetic refrigeration materials. Therefore, it is necessary to study the effect of MAO coatings with different thicknesses and compositions on thermal conductivity.
- Due to factors such as adjusting the phase transition temperature and reducing hysteresis loss, the La(Fe, Si)13 series alloys has multiple different compositions. Element doping and introduction of interstitial atoms are common methods and the surface and grain morphology change accordingly. The performance differences of MAO coatings generated on different La(Fe, Si)13 alloys deserve more extensive research.
After reading your review comments, we have discovered more points worth studying:
- As mentioned in review comment 3: “How does the thickness of the MAO coating influence the corrosion resistance, micromorphology, and magnetocaloric effect? Is there an optimal thickness range that balances these properties effectively?”
- How to precisely control the coating thickness by adjusting the electrolyte and other methods?
- The durability and stability of the MAO coating must be verified. (Different magnetic field intensity, temperatures, times, heat exchange media…)
Finally, please allow us to express our gratitude and respect to reviewer. Your questions are very professional and contains many aspects that can inspire our subsequent study. Because our research on this topic has just begun, there are many issues that we have not considered, and there are still many areas that need to be explored. Your review comments have largely guided our next steps of work. Once again, we would like to express our heartfelt gratitude!
Reviewer 3 Report
Comments and Suggestions for Authors
The article entitled "Preparation and performance of micro-arc oxidation coatings for corrosion protection of the LaFe11.6Si1.4 alloy" has been submitted for publication in the journal Materials (MDPI).
The research with the micro-arc oxidation of an innovative alloy combining lanthanum, iron, and silicon for corrosion protection. The effect of three different solutions is explored. These three solutions contain 2, 3, or 4 different electrolytes. The obtained surface layers are characterized by XRD, SEM, EDS, porosity measurements, adhesion measurements, potentiodynamic polarization curves, and DSC.
The results show improved corrosion behaviors due to the MAO surface layer.
In my opinion, this article contains interesting results that deserve to be published. However, the article lacks scientific discussion to highlight more the novelty of the work and to explain the mechanisms involved in the observed improvements.
For these reasons, I recommend major corrections of the manuscript according to the following points:
- The novelty of the work should be highlighted more. Why are the results interesting in comparison to the scientific literature dealing with similar topics?
- the article describes the impact of the electrolyte on the performance of the surface layer. However, there is no discussion dealing with the reason why this surface layer is better than the others. Is it just because the amount of alumina is higher? What is the mechanism of this improvement?
- In the whole manuscript, the XRD characterizations should mentioned with the word "pattern" instead of "spectra". In addition, the differences between the XRD patterns are not obvious. Several signals are barely distinguishable from the background. For example, that seems difficult to be affirmative with the presence of Fe2O3, FePO4, and FeAl2O4. Is there any alternative solution to confirm these claims?
- the description of XRD indicates variations in the intensity of some peaks. What is the reason for such observations? What is the mechanism that produces this effect? Is it good or bad for the expected applications? Was it already observed in the literature or is it the first time? More discussion is necessary.
- at the beginning of section 3, "Notably, owing to the layered porous structure of the coating, X-rays can penetrate the coating onto the substrate, resulting in a strong diffraction peak of Fe in the substrate. This also indicates that the Fe on the substrate participates in the MAO reaction". How do we know that the coating has a "layered porous structure" at this point? The reason for the incidence on the intensity of the diffraction peak should be explained and supported by relevant references. Similarly, why does this observation justify some diffraction of iron from the substrate?
- in Table 2, only one decimal digit should be used for these results. Are these results weight percents or atomic percents?
- unprecise words like "approximately", and "relatively small"… must be avoided, they are not scientifically accurate.
- the table with the corrosion results must be named "Table 3". The potential value must be indicated relatively to a reference electrode. The unit of the corrosion rate is generally indicated "per year" with the letter "y" instead of "a".
- the unit Pascal must be written with a capital letter "Pa".
Comments on the Quality of English LanguageThe English language is fine, only minor corrections are needed.
Author Response
Dear Editors and Reviewers:
Thank you for your letter and for the reviewers’ comments concerning our manuscript. All the comments are valuable and very helpful for revising and improving our paper, as well as the important guiding significance to our researches. We have addressed all the comments carefully and have made answers, supplements and corrections. In addition, some errors in the references have been corrected. All revisions are highlighted in red font in the manuscript. The responds to the reviewer’s comments are as following:
Comment 1: The novelty of the work should be highlighted more. Why are the results interesting in comparison to the scientific literature dealing with similar topics?
Response: Indeed, as stated in this comment, the purpose and novelty of this work were not expressed clearly. In fact, the reason we attempt to prepare the MAO coatings on rare earth iron based alloys is the characteristics of this technology are highly suitable for the corrosion protection requirements of La(Fe, Si)13 magnetic refrigeration materials. But the significance of this work was not clearly stated in manuscript. Therefore, in the introduction section of the revised manuscript, corresponding information have been added. Thank you for this important comment!
Comment 2: The article describes the impact of the electrolyte on the performance of the surface layer. However, there is no discussion dealing with the reason why this surface layer is better than the others. Is it just because the amount of alumina is higher? What is the mechanism of this improvement?
Response: Thank you for this comment. Based on experimental analysis, we believe that the better performance of the coating is due to two factors. On the one hand, the addition of Na2CO3 and Na2B4O7 improves the discharge mechanism of micro-arc oxidation. This factor changes the formation mechanism of the surface layer, leading to the disappearance of cracks, the sealing of most discharge holes, resulting in a smoother and more uniform surface layer. On the other hand, the addition of Na2CO3 changes the solution environment, and the addition of Na2B4O7 promoted the decomposition of iron oxide, making it more conducive to the generation of alumina, resulting in a higher alumina content in the coating. This factor greatly improves the corrosion resistance of the coating. The combined effect of the above two factors improves the performance of the surface layer. The logic was not expressed clearly in the original manuscript, so we made targeted modifications to the discussion of Fig. 3 and Table 2 in the revision.
Comment 3: In the whole manuscript, the XRD characterizations should mentioned with the word "pattern" instead of "spectra". In addition, the differences between the XRD patterns are not obvious. Several signals are barely distinguishable from the background. For example, that seems difficult to be affirmative with the presence of Fe2O3, FePO4, and FeAl2O4. Is there any alternative solution to confirm these claims?
Response: Thank you for this comment. Firstly, we have corrected the incorrect expression of "spectrum". Secondly, we conducted another XRD test on the sample to address the issue of poor signal. Additionally, due to the alteration of the electrolyte, there is a gradual decomposition of iron oxide, resulting in the diminished prominence of the Fe2O3 diffraction peak in M3. Meanwhile the XPS and EDS experimental results support the existence of Fe2O3, FePO4, and FeAl2O4 from another perspective. The mutual verification of above experimental results has convinced us presence of Fe2O3, FePO4, and FeAl2O4 in the samples.
Comment 4: The description of XRD indicates variations in the intensity of some peaks. What is the reason for such observations? What is the mechanism that produces this effect? Is it good or bad for the expected applications? Was it already observed in the literature or is it the first time? More discussion is necessary.
Response: Thank you for asking these questions. After analysis, it is believed that the reason for this phenomenon is: on the one hand, the addition of Na2CO3 promoted the growth of alumina in the coating, and on the other hand, the addition of Na2B4O7 promoted the decomposition of iron oxide. These two processes caused changes in peak intensity in the XRD pattern. Due to the fact that the generation of alumina is beneficial for improving the corrosion resistance of coatings, this effect is good for the practical application of La(Fe, Si)13 magnetic refrigeration materials. Previous articles have reported this phenomenon on other substrate materials. Relevant references have been added at the corresponding positions in revised manuscript. In addition, the XRD section did not discuss in detail the reasons for the increase and decrease of compounds such as iron oxide and aluminum oxide, mainly because we feel that there is still too little experimental evidence. In the latter half of the paper, combined with experimental data such as SEM and EDS, the mechanism of this phenomenon has been analyzed and elaborated. Meanwhile, this comment also reminds us that our discussion was not in-depth enough. Therefore, combined with this and the previous review comments, we added corresponding explanations after Figure 3 and Table 2.
Comment 5: At the beginning of section 3, "Notably, owing to the layered porous structure of the coating, X-rays can penetrate the coating onto the substrate, resulting in a strong diffraction peak of Fe in the substrate. This also indicates that the Fe on the substrate participates in the MAO reaction". How do we know that the coating has a "layered porous structure" at this point? The reason for the incidence on the intensity of the diffraction peak should be explained and supported by relevant references. Similarly, why does this observation justify some diffraction of iron from the substrate?
Response: Thank you very much for this comment. Due to this opinion, we discovered an error in the text: it should not be the diffraction of Fe, but rather the diffraction of Fe2O3. This is because there is a dense Fe2O3 layer between the substrate and the coating, as can be seen in Fig. 4. This error has been corrected in the revised manuscript. For the layered porous structure, the reason for its formation is that during the process of micro arc oxidation, some residual discharge channels change from through holes to blind holes. Meanwhile, with the continuous change of discharge intensity, the growth rate of the coating also changes, resulting in a layered structure. The combination of two factors results in the formation of a layered porous structure in the coating. The relevant literature describing this mechanism has been added to the corresponding position.
Comment 6: In Table 2, only one decimal digit should be used for these results. Are these results weight percents or atomic percents?
Response: Thank you very much for pointing out this irregularity. The results in Table 2 are atomic percentages. The numbers and titles in the Table 2 have been modified and supplemented respectively.
Comment 7: Unprecise words like "approximately", and "relatively small"… must be avoided, they are not scientifically accurate.
Response: Thank you for the professional suggestions! We have reviewed the manuscript and made corrections to such unprecise statements.
Comment 8: The table with the corrosion results must be named "Table 3". The potential value must be indicated relatively to a reference electrode. The unit of the corrosion rate is generally indicated "per year" with the letter "y" instead of "a".
Response: Through this review comment, we found that we mistakenly labeled Table 3 as Table 2. We have corrected this error and sincerely apologize for this low-level mistake. In accordance with this comment, the unit of the corrosion rate has been modified, and descriptions of some parameters in this table such as Icorr and Ecorr have been also marked. Also, we have provided an explanation of the electrode system in this section. Thank you for these professional suggestions!
Comment 9: The unit Pascal must be written with a capital letter "Pa".
Response: Thank you for your careful reading of our manuscript! We found this mistake in the conclusion section and made correction.
Round 2
Reviewer 3 Report
Comments and Suggestions for Authors
The authors have appropriately corrected their manuscript according to my comments.
I recommend the article materials-2903307 to be accepted for publication in the journal Materials.
Comments on the Quality of English LanguageEnglish language is fine.